# Effects of Enrichment Type, Presentation and Social Status on Enrichment Use and Behaviour of Sows with Electronic Sow Feeding

**DOI:** 10.3390/ani9060369

**Published:** 2019-06-18

**Authors:** Cyril Roy, Lindsey Lippens, Victoria Kyeiwaa, Yolande M. Seddon, Laurie M. Connor, Jennifer A. Brown

**Affiliations:** 1Prairie Swine Centre, Box 21057, 2105-8th Street East, Saskatoon, SK S7H 5N9, Canada; Cyril.Roy@usask.ca (C.R.); Victoria.Kyeiwaa@usask.ca (V.K.); 2Department of Animal Science, Faculty of Agricultural and Food Sciences, 201-12 Dafoe Road, University of Manitoba, Winnipeg, MB R3T 2N2, Canada; Lindsey_2310@hotmail.com (L.L.); Laurie.Connor@umanitoba.ca (L.M.C.); 3Western College of Veterinary Medicine, University of Saskatchewan, 52 Campus Drive, Saskatoon, SK S7N 5B4, Canada; yolande.seddon@usask.ca

**Keywords:** environmental enrichment, social status, sows, aggression, habituation

## Abstract

**Simple Summary:**

Many North American pork producers are transitioning to group housing systems for gestating sows and are looking to provide animals with environmental enrichment. Because straw poses biosecurity and manure management concerns it is important to identify alternative enrichments which can benefit sow welfare. The effects of four enrichment treatments were studied: (1) Constant: constant provision of wood on chain; (2) Rotate: rotation of three enrichments (rope, straw and wood on chain); (3) Stimulus: rotation of three enrichments with an associative stimulus (bell or whistle); and (4) Control: no enrichment. Contacts with enrichment and time spent in different postures were measured using scan sampling for all sows. Skin lesions were scored and cortisol was measured in saliva samples in a subset of dominant and subordinate sows. Sows spent more time contacting enrichment in Rotate and Stimulus treatments than Constant, particularly when straw was provided. Subordinate sows spent more time near enrichments, and more time standing than dominant sows. Subordinate sows also received more skin lesions and had higher salivary cortisol concentrations than dominants. We conclude that enrichments are valued by sows, and that the right amount is needed to minimize competition over access. Additional work is needed on rotational schedules and determining appropriate levels of enrichment for sows.

**Abstract:**

The goal of this study was to identify practical enrichments for sows in partially or fully slatted pen systems. Four treatments were applied: (1) Constant: constant provision of wood on chain; (2) Rotate: rotation of rope, straw and wood enrichments; (3) Stimulus: rotation of enrichments (as in Rotate) with an associative stimulus (bell or whistle); and (4) Control: no enrichment, with each treatment lasting 12 days. Six groups of 20 ± 2 sows were studied from weeks 6 to 14 of gestation in pens with one electronic sow feeder. Each group received all treatments in random order. Six focal animals (3 dominant and 3 subordinate) were selected per pen using a feed competition test. Digital photos were collected at 10 min intervals for 8 h (between 8 a.m. and 4 p.m.) on 4 days/treatment (d 1, 8, 10 and 12) to record interactions with enrichment. Skin lesions were assessed on days 1 and 12, and saliva cortisol samples collected in weeks 6, 10 and 14 of gestation on focal pigs. Sows spent more time in contact with enrichments in Rotate and Stimulus treatments than Constant. Enrichment treatments did not influence lesion scores. Subordinate sows spent more time standing and near enrichments than dominants. Subordinate sows also received more skin lesions and had higher salivary cortisol concentrations than dominants. These results indicate that access to enrichment is valued by sows but can result in greater aggression directed towards subordinates.

## 1. Introduction

As gestation stalls are replaced with group housing in North America, there are increasing opportunities to provide enrichment to sows and potential to improve the health and welfare of pigs [1]. Sows managed in group housing may benefit from social enrichment (interaction with other sows), if appropriate space allowance and feeding management are provided. However, particularly for subordinate group members, there are risks associated with group housing such as increased aggression [2,3] and potential health risks such as lameness, injury and abortion [4]. 

Scientific evidence on environmental enrichment for pigs indicates that it has the potential to increase social interactions, reduce stereotypies and increase general activity [5,6]. While research on enrichment use in sows is limited, some positive effects associated with enrichment in growing pigs include increased ability to adapt to novel situations, reduced incidence of negative behaviours such as belly nosing, tail and ear biting, reduced fearfulness and improved learning capabilities [1,7,8,9,10,11,12]. Extrapolating from these studies, it is hypothesised that some of the negative aspects of housing sows in groups, such as aggression and oral stereotypies can be mitigated by providing enrichment. 

In North America, partially slatted or fully slatted floors with liquid manure systems are common and limit the use of substrate enrichments due to the potential for blocking the slurry system [10]. Other concerns related to substrate provision include cost and biosecurity risk. Point source object enrichment (enrichment provided at a fixed position in the pen) may be more suitable for slatted or partially slated flooring systems. Some point source enrichments that have been previously studied include a rubber bar, rubber ball, ribbon, rope, garden hose, wood, chains [8,13,14] straw or hay [1,7] and sexual pheromones [15]. 

Many factors are known to affect enrichment use by pigs. For enrichment to be effective, it should be kept clean and not soiled with feces, easily accessible, attractive to the animals, deformable, destructible, and immobile so that the animals are able to hold and manipulate it [13]. Research has shown that presenting enrichment materials in different ways also helps to maintain their attractiveness [16]. In piglets, announcing the provision of enrichment (by using a sound stimulus) resulted in a significant increase in play behaviour and reductions in aggression and lesion scores following weaning [9]. Social status within the group can also affect activity and the behaviour of sows. If enrichment is valued and access is limited then dominant sows may displace subordinates and limit access to enrichment [17]. Additional research is needed to understand how these factors affect enrichment usage in sows. 

Habituation can also reduce the effectiveness of enrichments, particularly when point source physical enrichments are provided [8,18]. When the same enrichment material is used over time, animals lose interest due to habituation [19]. The initial response and interaction with point source objects is generally high (within the first 24 h), and then declines over subsequent days or weeks [15]. However, habituation to a point source object is also dependent on its properties and the method of presentation [15]. For example, Van de Weerd et al. [18] observed that hanging sisal rope had a higher initial response and interaction frequency on day 1 when compared with loose concrete blocks or a rubber boot provided on the same day. This highlights the importance of understanding how different presentation methods affect enrichment use. Previous studies also indicate that pigs are attracted to enrichments that are destructible, deformable and chewable [8,13,18] indicating that the properties of the enrichment can also impact their use. Therefore, there is a need to understand how different types of enrichment and presentation methods influence habituation and enrichment usage by sows in group housing.

In this study, three enrichment treatments were provided to group housed sows and were compared against a control treatment involving no enrichment. Three enrichment materials, including cotton rope, wood and straw were selected for the trial. Our objective was to examine how enrichment type, method of presentation and the social rank of sows affected enrichment use and sow behaviour. We hypothesized that frequent rotation of enrichments would increase enrichment use and reduce habituation, and that the use of an associative stimulus would increase the initial response to enrichment provision. Furthermore, we hypothesized that enrichment provision would reduce social stress and aggression among sows.

## 2. Materials and Methods 

The research was conducted at the research barn associated with Glen Lea Research Centre (University of Manitoba, Winnipeg, MB, Canada). The experiment was approved by University of Manitoba Committee on Animal Care (AUP# F2014-031/1/2) and adhered to the Canadian Council on Animal Care guidelines for humane animal use.

### 2.1. Animals and Housing

A total of 120 sows (20 ± 2 sows in 6 replicates, Genesus genetics, Oakville, Manitoba, Canada) were studied. Sows were housed in groups on partially slatted concrete floors. Trials were conducted in four gestation pens of two designs, with an average pen area of 61.2 m^2^ (3 m^2^/sow). Two pens had 25% solid floor area and two pens had 43% solid floor area, with the remaining portion being slatted (Figure 1). Feed was supplied using Electronic Sow Feeding (ESF; Nedap Velos, Nedap Livestock Management, Groenlo, The Netherlands). Sows were artificially inseminated with pooled semen, and remained in groups of 3–5 sows with feeding stalls for four to five weeks before being mixed into gestation pens after confirmation of pregnancy. Sows entered the study following mixing into gestation pens in week 4–5 of gestation. 

A sub-sample of 3 dominant and 3 subordinate sows per group was identified and marked in week 1, based on feed competition trials, following procedures adapted from Anderson et al. [19]. Feed competition testing was performed in the pen on days 3–5 after mixing, after the initial aggression associated with mixing had resolved. On the afternoon of the third day after mixing, the solid floor area was scraped and sows were given 4 kg of feed poured onto the floor in two lines (2 kg per line, approx. 1 m long in 2 lines, spaced at least 2 m apart). Once the feed was placed, sows were allowed to compete for floor feed. This procedure was repeated on days 4 and 5 after mixing. On the third day of the feed competition trial, sows were observed and three sows which gained first access to feed were identified as ‘dominant’ (Dom), and three sows that refrained from feed competition and/or were driven away were identified as ‘subordinate’ (Sub). 

### 2.2. Treatments

Four enrichment treatments were provided to each group of sows (pen). Each treatment was provided for 12 days, with treatment order randomized and a two-day interval between treatments. Treatments consisted of: Constant provision of one type of enrichment—wood on chains, 3 per pen (Constant),Rotation of three enrichments—rope, straw, wood on chain (Figure 2), 3 per pen (Rotate),Rotation of three enrichments (as described for Rotate) with an associative stimulus used to signal the arrival of enrichment (Stimulus). The associative stimulus used was a bell or whistle (duration: 2 s), and was switched half-way through the study so that any sows that returned to the study (in their next gestation) would not be familiar with the stimulus, andNo enrichment (Control).

In Rotate and Stimulus treatments, the enrichments were rotated three times per week, with rope on Mondays, straw on Wednesdays and wood on Fridays (see timeline, Figure 3). For the Stimulus treatment, a bell or whistle was used immediately before providing enrichment as an associative stimulus. Wood on chain and rope enrichments were hung over the slatted area of each pen using carabiner clips to attach the enrichment to chains attached to the ceiling using an eye bolt. The rope length was 1.2 m (including a 15 cm tassel at the end), using three-stranded cotton rope 19 mm in diameter and suspended 20 cm above floor level. Straw was provided on the solid floor area such that there was 300 g per sow (8.4 kg in total). Any remaining straw was removed before giving the next enrichment, but most of the material was consumed by sows. The wood enrichment was made of softwood, 5 × 10 cm and 1.2 m in length and hung from the chain in such a way that it rested on the floor at 45° angle.

### 2.3. Data Collection

The study began the week that sows were moved to gestation pens (4–5 weeks of gestation). The body weight and parity of individual sows was recorded as sows entered group gestation pens. 

Enrichment use was studied by mounting one camera over each pen and recording sow activity around the enrichments. Digital photos were taken at 10 min intervals over 8 h per day (8 a.m.–4 p.m.) on days 1, 8, 10 and 12 of each treatment (approximately 48 photos per day and 192 over the four days of observation). Photos were transcribed by one trained observer, who determined the location and posture of all sows visible in the enrichment area. All sows that were clearly visible were observed and the number of sows standing, lying or sitting at each time point was recorded. Table 1 shows an ethogram of behaviour categories recorded. For Rotation and Stimulus treatments, the recordings coincided with the first 8 h following provision of new enrichments. In the Stimulus treatment the “stimulus” (2 s, bell or whistle) was given as soon as cameras were started, and was followed by enrichment provision. Enrichment use was studied by transcribing the photos in two datasheets: First as a whole group and second recording only the 6 selected Dom and Sub sows. Dom and Sub sows were identified by blue (Dom) or red (Sub) spray markings.

#### 2.3.1. Skin Lesion Assessment 

Lesion scores were used to evaluate levels of aggression in pigs, using methods adapted from Hodgkiss et al. [20]. For each treatment period, skin lesion scores were assessed on day 1 (before enrichment was provided) and on day 12. Lesion scores ranged from 0 (no injury) to 3 (severe injury), and were assessed on 11 regions (head, ear, neck, shoulder, top of back, tail, vulva, hind leg, side, udder and front leg) on both the right and left sides of the body (Figure 4). The scoring accounted for fresh injuries only, following the description; 0 = No injury (skin unmarked: no evidence of injury), 1 = Slight injury (<5 superficial wounds), 2 = Obvious injury (5–10 superficial wounds and/or <3 deep wounds), 3 = Severe injury (>10 superficial wounds, and/or >3 deep wounds). 

#### 2.3.2. Cortisol in Saliva 

Saliva samples were collected from the three Dom and three Sub sows per group at three time points: (1) The end of week 1 after application of first treatment; (2) day 11 of the second treatment (i.e., week 9 of gestation); and (3) on completion of the final treatment (week 14). Saliva samples were collected between 8 and 9 a.m. each day to control for diurnal variation in cortisol levels. Saliva was collected by allowing sows to chew on large cotton buds wrapped on a metal support until the bud was thoroughly moistened, about 30 to 60 s per sample. The moistened buds were placed in 15 mL centrifuge tubes (Fisher Scientific, Ottawa, ON, Canada) and centrifuged immediately for 15 min at 830× *g* (Beckman TJ-6 Centrifuge, Beckman Coulter, Mississauga, ON, Canada) to remove mucins and other particulate matter. Saliva samples were transferred to labeled storage tubes using disposable pipettes and stored at −20 °C until analysis. 

The cortisol concentration in saliva was determined using the Salimetrics^®^ Cortisol Enzyme Immunoassay Kits (Salimetrics, State College, PA, USA) for research in humans and animals. The Kit is a competitive immunoassay designed for quantitative measurement of salivary cortisol using a 96-well ELISA plate with spectrophotometric detection at 450 nm. The manufacturer’s instructions were followed and each 96-well plate contained six standards, one zero and one nonspecific binding sample, two controls and 38 samples, all were run in duplicate. Intra-assay precision estimates on low and high standards (*n* = 20) gave values of 1.14 ± 0.05 and 0.16 ± 0.01 µg/dL (mean ± SD) and CV’s of 4% and 5%, respectively. Inter-assay precision estimates on low and high standards (*n* = 20) gave values of 1.14 ± 0.05 and 0.18 ± 0.01 µg/dL (mean ± SD) and CV’s of 4% and 9%, respectively. 

### 2.4. Statistical Analysis

Measurements were done on 6 pens of sows, with four enrichment presentations (treatments) applied to each group in random sequence. Each treatment lasted 12 days, with sows’ behaviour observed on days 1, 8, 10 and 12. Therefore, for all group behavior observations, *n* = 6 groups × 4 treatments × 4 observation days = 96 observations were analyzed using repeated measures models in SAS 9.3 (SAS Institute Inc., Cary, NC, USA). 

*Data manipulation:* Pen was the experimental unit when analyzing outcome variables associated with enrichment use (location and postures) both for group level and focal sow photo scan observations. The proportion of time in each behaviour (defined in Table 1) was calculated as the proportions of scans (photos) where the behaviour was observed out of the total number of scans (photos) recorded during the 8 h observation period (48 scans per day). The number of sows performing each behaviour was calculated by the average proportion of sows that performed the behaviour when the behaviour was observed.

Sow was the experimental unit for lesion scores and cortisol analysis. For lesion scores, a combined skin lesion score was created by summing the lesion scores for all body regions. The effects of different enrichment presentations and treatment day (day 1 and day 11) were assessed using the total body score. To study lesion scores in different body regions, the 11 regions (Figure 4) were condensed in to four regions (head, shoulder, side and hind) and assessed for associations with independent variables enrichment treatment, social status and parity. Parity was categorised in three groups: Parity code 1 (parities 1 and 2), parity code 2 (parity 3) and parity code 3 (parities 4 to 7). 

*Data Analysis:* Location and postural behavioural data were analysed using GLMM with beta regression models using SAS 9.3 to account for proportional data. Model fit was assessed by plotting the residuals. Fixed effects in the group behaviour model included enrichment treatment, day of treatment (day 1, 8, 10 and 12) and their interaction. The effects of social status on sow behaviour was studied in a similar model with social status added as a fixed effect and including two-way interactions of social status and treatment and social status and day of treatment. Sow group (replicate) and day of treatment were used as random effects to account for the correlation associated with repeated measurement of the same animals. Day of gestation was controlled for as each group (pen of sows) received all treatments, and the order of enrichment treatments was randomized. The significance level was set at *p* < 0.05.

Proc Mixed was used to assess the association between cortisol concentration and fixed factors including social status and parity category, and sow group (replicate) was the random effect. A similar model was used for lesion scores with measures for day 1 and 12 observations being analysed separately and individual sow ID was added as a repeated measure. 

## 3. Results

### 3.1. Group Level Observations

#### 3.1.1. Enrichment Use at Group Level

Significantly more sows were observed contacting enrichments in the Rotate and Stimulus treatments, compared to Constant treatment (Table 2). Day of observation also had significant effect on the number of sows contacting enrichment (Figure 5). On day 10, when straw was provided in the Rotate and Stimulus treatments, sows contacted enrichment more frequently than when rope (days 1 and 8) or wood (day 12) were provided (*n* = 6, F (3, 62), *p* = 0.001). Also, more sows contacted the straw than when wood (day 1, 8, 10 and 12) was provided in the Constant treatment (Figure 5). No significant interactions were found between enrichment treatment and day of observation.

#### 3.1.2. Behaviour at Group Level

When sow postures (standing, sitting and lying) were studied at group level, sows were observed to spend more time sitting in the Control (no enrichments), Constant and Rotate treatments compared to the Stimulus treatment (*n* = 6, F (3, 89) = 2.99, *p* = 0.035, Table 3). There was a tendency for more sows to stand in Rotate and Stimulus than in Control or Constant treatments (*p =* 0.081, Table 3).

### 3.2. Focal Pig Observations

The parity distribution of Dom and Sub focal sows was: Dominants- parity 2: 39%; 3: 11%, 4: 17%, 5: 17%, 6: 11% and 7: 5% and Subordinates- Parity 1: 11%, 2: 28%, 3: 22%, 4: 22% and 5: 16% (18 Dom and 18 Sub sows in total). Parity had no effect on any measures in focal sows.

#### 3.2.1. Enrichment Use by Focal Sows

The effect of social status was studied by observing a sample of six focal sows (3 Dom and 3 Sub) per group of 20 ± 2 sows. Sub sows spent more time close to enrichments (<1 m) than Dom (*p* = 0.001), and also more time >1 m from enrichments than Dom sows (*p* = 0.001, Table 4). 

#### 3.2.2. Behaviour of Focal Sows

Social status had a significant effect on standing behaviour, with Sub sows spending more time standing and higher number of sows performing this activity when observed (Table 5). No interactions were found among social status, treatment or day.

#### 3.2.3. Skin Lesions

Sub sows received more lesions on the shoulder area than Dom sows (*p* < 0.001, Table 6), and had numerically higher lesion scores on all body regions. Social status had no significant effect on head, side or hind lesions (data not shown). There were significant differences in lesions scores among treatments (Table 7). On treatment day 1, sows in the Stimulus treatment had significantly higher lesion scores than sows in all other treatments. However, because day 1 lesions were assessed as a baseline, before enrichment was given, these differences cannot be attributed to the enrichment treatment. Sows may have fought more during the two-day interval between treatments when no enrichment was provided. The absence of significant treatment effects on day 12 lesion scores is arguably a more important result. No interactions were found between social status and treatment for lesion scores.

#### 3.2.4. Cortisol in Focal Sows

Social status had a significant effect on the cortisol levels of sows, with Sub sows having significantly higher cortisol concentrations than Dom sows (Table 8). Sampling time point also had a significant effect, with cortisol levels in late gestation being significantly higher than in early or mid-gestation. 

## 4. Discussion

This study compared the effects of different types of enrichments and presentation methods on sow behavior, and the influence of social status on enrichment interaction and behaviour. Two enrichment presentation methods (Rotate and Stimulus) were used to compare the effects of different enrichment materials (straw, rope and wood) on the time spent interacting with enrichment and sows’ proximity to enrichment. The results indicate that sows interacted with straw, rope and wood in decreasing order. This is in agreement with previous studies which found that straw produced a greater behavioural response compared to object enrichments [10,21,22]. Elmore et al. [23] observed that sows spent more time using a straw rack compared to mushroom compost, suggesting a preference for consumable enrichments over non-consumable enrichments. Straw was included in the present study as a “positive control” for comparison with object enrichments. As predicted, more sows were observed contacting enrichment when straw was provided. The greater use of straw enrichment in the current study was also influenced by the fact that straw was provided on the floor, in a more diffuse manner than hanging objects, allowing more sows to interact with the material at once. This study also showed that providing 300 g of straw per sow in partially slatted pens did not cause any blockage of the liquid manure system. Most of the material was consumed by sows and only a small amount entered the pits. Limitations to the study design include the fact that the rotation of rope, straw and wood was always done in the same order, which could have influenced the pigs’ behavior, and that the two pen designs used may have produced different effects but were not included in the model due to low sample size.

Although rope and wood enrichments elicited less contact by sows than straw in this study, their usefulness as enrichment materials cannot be discounted. Previous studies have also reported ‘hanging rope’ as an effective enrichment strategy [10,14,17]. Multiple studies indicate that for pigs to show sustained interest in a particular enrichment, the enrichment should be ingestible, flexible and destructible [8,13,18]. Rope and wood satisfy at least two of the described characteristics (flexible and destructible) and hence can be good objects for provision of enrichment. However, the accessibility of enrichment material (ratio of enrichments to sows) is another factor that must be considered, as limited access to desirable forms of enrichment may increase competition and aggression. 

Pigs’ interest in enrichment can be sustained by changing objects or renewing substrates on a regular basis [8,24] thereby mitigating effects of habituation. We hypothesised that frequent rotation of the enrichments (as in the Rotate and Stimulus treatments) would maintain novelty (reduce habituation) and increase the activity level of sows. The results show that rotation of enrichment objects resulted in significantly more sows within 1M and contacting enrichments. It is likely that changing enrichments every three to four days was responsible for this result. In a study of growing pigs, Van de Perre et al. [8] tested a sequence of enrichment objects by rotating seven different enrichment objects every week throughout the growing period and found a significant reduction in pen mate biting behaviour and wounds compared to presentation of a chain enrichment only. Pigs’ interaction with enrichment was greatest on day 1, and declined significantly by day 5 of each rotation [8]. Grifford et al. [25] studied the ability of pigs to recognize objects following different exposure periods (10 min vs. 2 days) and delays before re-exposure. Based on the results they suggested restricting object exposure to less than two days in rotated objects, and delaying re-exposure to the same object by more than one week to enhance the exploratory value of enrichment. Since both of these studies used growing pigs, generalizing the results to sow behaviour may not be appropriate. Since novelty played a role in increasing the interest of sows in this study, further studies should examine how different rotation frequencies and delays to re-exposure of a particular enrichment can influence sow behaviour, as well as the optimal ratio of enrichments to sows. 

Pairing enrichment with another stimulus has been used to increase the value of enrichments by some researchers. For example, the study by Dudink et al. [9] used an associative stimulus in conjunction with enrichment provision, which resulted in anticipatory excitement and increased play behaviour in young pigs. By presenting enrichments in conjunction with an associative sound stimulus in the current study, we hypothesized that the stimulus would increase the initial response of sows to the delivery of new enrichments. However, no evidence of an increased initial response was found. Douglas et al. [26], paired enrichments with an auditory stimulus in gilts and measured an improvement in their cognitive performance. In both the Dudink et al. [9] and Douglas et al. [26] studies, pigs were given straw in addition to food rewards in the form of mixed seeds or sliced apple, respectively, after the sound cues. The studies also used an intensive training period for pigs to learn the association between the rewards and sound cues. There was 30 s delay after the sound cue before rewards were given and the sequence of the rewards and auditory cues was randomised. Our goal in this study was to simulate enrichment provision as it would be implemented in commercial practice, so no training phase was used. Because we did not use a training phase, it is unclear whether sows made a clear association between the sound cue and enrichment provision. Future studies should include a training phase so that the association is learned, and should also recognise the potential for an associative stimulus to increase competition and aggression if access to enrichment is limited. 

One of the goals of providing enrichment is to increase the frequency and duration of normal behaviours, such as increasing activity as represented by standing behaviour [27,28,29]. If enrichment treatments are effective, we would expect to see a higher level of activity when enrichment is presented than in a barren pen environment. In terms of sows’ postures, rotating enrichments (Rotate and Stimulus treatments) resulted in more sows standing, and the Stimulus treatment reduced the duration of sitting, suggesting that sow activity increased when enrichments were rotated. Gestating sows are generally inactive, spending roughly 80% of their time lying [30]. However, a moderate level of activity should improve fitness and be beneficial, as shown in studies comparing stall and group housed sows [31]. Further research should explore potential benefits of increased activity and “appropriate” levels of activity for gestating sows. 

Regarding the effects of social status on enrichment interaction, we hypothesised that if enrichment was considered a valuable resource, dominant sows would have greater access to enrichment or access it at preferred times, compared to subordinates. However, the results show that, sows gained access to enrichments regardless of social status, and in fact Sub sows spent more time near enrichments than Dom sows. One explanation for this may be that Dom sows may have been more preoccupied with guarding the ESF feeder, as the feeder is clearly an important resource within the pen. Furthermore, because social status was determined based on a feed competition test, it may be that differences in social status are confounded with feeding motivation. Elmore et al. [17] found no significant effect of social status on stall-housed gestating sows’ motivation to access an enriched group pen. In their study, both dominant and subordinate sows showed similar levels of an operant response and similar latency to press a panel to access enrichment objects. However, once the sows were released into a group setting, social status had a significant effect on enrichment use, with dominant sows spending more time interacting with the objects than subordinates. 

Elmore et al. [17] observed significant effects of social status on active behaviour such as standing and inactive behaviours, with dominant sows being more active and standing more compared to subordinate sows. However, in this study, subordinate sows spent more time standing than dominants. Because behaviour observations were collected using a camera focused on the enrichments, the behaviour of Dom sows in other pen areas was not recorded. As suggested previously, Dom sows appear to have been more focused on the feeder area, which may explain the greater enrichment use and increased activity level of subordinate sows in this study. It also suggests that either there was sufficient availability of enrichment objects (three per 20 sows) to reduce competition, or that dominant sows did not view the enrichments as highly valuable. 

Lesion scores have been used in previous studies to determine the aggression level of pigs [20]. Hodgkiss et al. [20] and Stukenborg et al. [32] demonstrated that the severity, duration and frequency of overt aggression or fighting can be estimated by looking at the number of scratches or injuries on pigs, especially injuries to the anterior region of their body. However, aggression is also influenced by factors such as the familiarity of pigs, space allowance, group size and composition, pen design, time of day, food and bedding [33]. According to these authors, aggression or its absence in the social environment is affected largely by the degree of competition over resources. For example, an increase in aggression was found when group-housed grower-finisher pigs were given limited access to a highly valued enrichment—straw [34]. We hypothesised that subordinate sows would receive more skin lesions due to aggression than dominant sows. Since the lesion results are consistent with that hypothesis, we conclude that subordinate sows experienced more aggression than dominants in this management system, however, it should be noted that the scores were consistently low. 

Subordinate sows in this study had significantly higher salivary cortisol concentrations compared to dominant sows. Because all sows received all enrichment treatments (in random order) and saliva samples were only collected at three time points, it was impossible to determine effects of the enrichment treatments on cortisol. Usually the social hierarchy is established in gestation sows within 24–48 h after mixing in a group housing type of management. Studies have found an increase in cortisol concentration shortly after mixing in group housing due to social aggression, and after a period of 48 h cortisol concentration returns to baseline [35]. There is evidence to indicate that if there is no competition for resources, then there should be no difference in cortisol concentration between subordinate and dominant sows after a week. O’Connell et al. [36] examined the effects of social status on the welfare of sows in dynamic groups. They found no difference in salivary cortisol concentration among subordinate and dominant sows at one week after mixing. However, if there are insufficient resources such as space, food or environmental enrichment for the group, then stress levels remain elevated [37] particularly in subordinate sows. This in combination with lesion results indicates that subordinate sows experienced greater stress compared to dominants in this management system. Another possible explanation may be that due to ESF feeding (here with a 9 a.m. reset time), the diurnal cortisol rhythm of subordinate sows may be shifted. Further research to determine whether the ESF system affects the circadian rhythm of dominant and subordinate sows differently would be of interest. All sows showed higher salivary cortisol concentrations at 14 weeks gestation than at five or nine weeks. The increase in cortisol levels during late gestation was expected as this is known to occur as sows (and other species) approach parturition [38,39,40]. 

## 5. Conclusions

The different enrichment materials (rope, wood and straw) and how they were presented to sows both had significant effects on the total number of sows contacting enrichment. Straw enrichment provided on the pen floor produced the greatest response to object enrichments, however, its presence only in the Rotate and Stimulus treatments represents a confounding effect in this study. Sows also made use of the rope and wood on chains enrichments, with wood being the least contacted. Regularly changing enrichment materials (as in Rotation and Stimulus treatments) increased sows’ response to enrichment. These findings demonstrate that novelty and the type of material provided play an important role in increasing attractiveness and sustaining sows interest in enrichment. Future research should explore the most effective rotation schedule and the optimal ratio of enrichments to sows.

Social hierarchy (dominant or subordinate) had a significant effect on enrichment use, with subordinate sows having greater interaction with enrichment. We suggest that this may be due to dominant sows valuing the ESF feeder more than enrichments. Subordinate sows in this study had increased skin lesions, as well as higher salivary cortisol concentrations suggesting that these individuals experienced greater stress than dominants in this management system. Longer duration studies are needed to better evaluate the interaction between social status and enrichment treatments.

## Figures and Tables

**Figure 1 animals-09-00369-f001:**
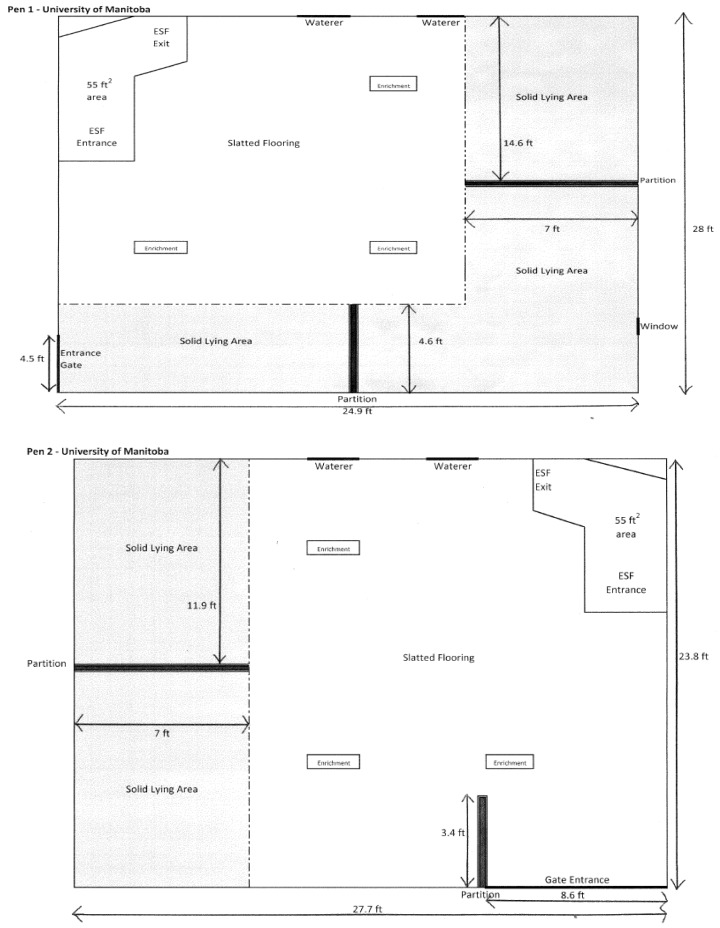
Pen designs used in the trial. Two of the pens had 43% solid floor area (**Pen 1**) and two pens had 25% solid floor area (**Pen 2**).

**Figure 2 animals-09-00369-f002:**
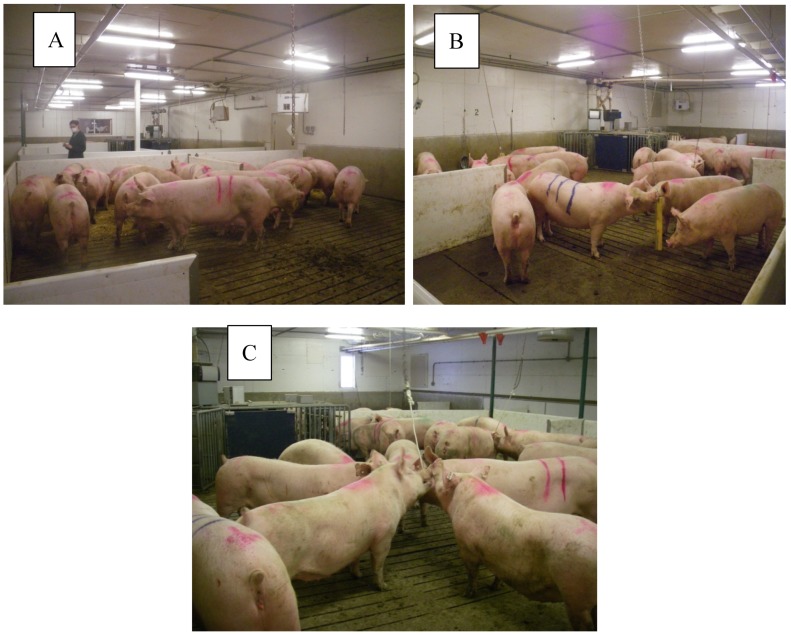
Photos illustrating presentation of straw (**A**), wood (**B**) and rope (**C**) enrichments to sows.

**Figure 3 animals-09-00369-f003:**
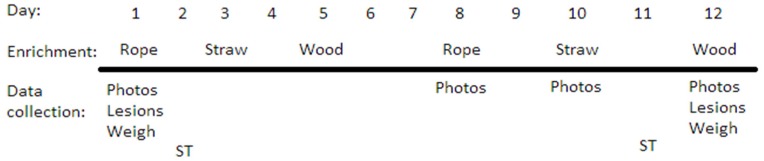
Timeline illustrating data collection and the rotation schedule for enrichments in Rotation and Stimulus treatments. Each treatment lasted 12 days, followed by two days off. Four treatments were provided consecutively to each pen group (*n* = 6) in random order over a period of eight weeks (beginning at 5–6 weeks and ending at 13–14 weeks of gestation).

**Figure 4 animals-09-00369-f004:**
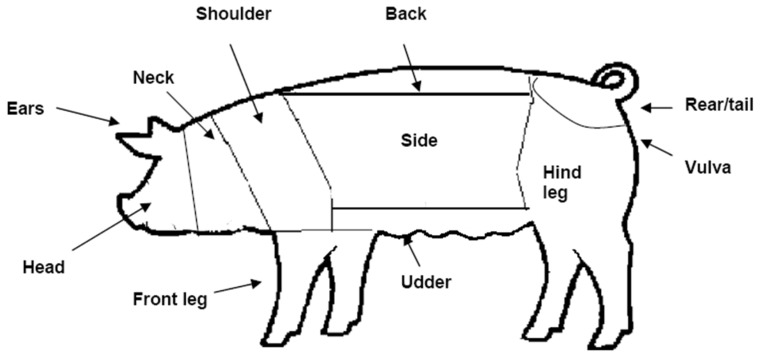
The animal’s body was divided in to 11 areas as illustrated. Each area on both sides was assessed for skin lesions (Score = 0 to 3).

**Figure 5 animals-09-00369-f005:**
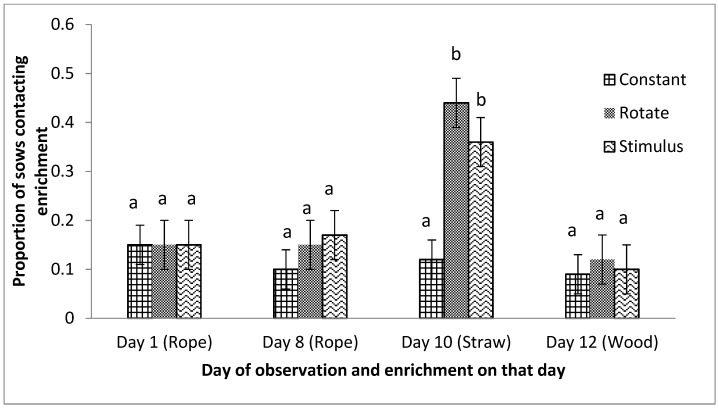
Proportion of sows contacting enrichments (rope, wood or straw) and day of observation (days 1, 8, 10 and 12) in the Constant, Rotate and Stimulus treatments. Note: sows in the Constant treatment received the wood enrichment on all days. Sows were observed during an 8 hour period (08:00–16:00 h) on four days (1, 8, 10 and 12) using time lapse photos at 10 min intervals. Bars without a common superscript are significantly different (*p* < 0.05). ^ab^ LSMeans with a different superscript are significantly different (*p* < 0.05).

**Table 1 animals-09-00369-t001:** Ethogram used to identify sow posture and location relative to enrichment.

Location/Posture	Definition of Location or Posture
In contact	Sow’s snout is in contact with enrichment. For straw enrichment, the sow should be standing on or facing the straw, and appears to be contacting the material with her snout.
Less than 1 m	Sow is not in contact with enrichment, but head is within approximately 1 m of enrichment. For straw enrichment, the sow is lying or standing in proximity to the straw (head is <1 m from straw)
Greater than 1 m	Sow’s head is greater than 1 m from enrichment. Includes all visible sows in the photo that are not in contact or <1 m.
Standing	Sow is upright on four legs (not sitting or lying). If the sow is in the process of lying down, however still upright and rear end is supported on hind legs, she is considered standing.
Sitting	Hind end is in contact with the floor; front end is raised and supported by front legs.
Lying	Sow is lying (ventral or lateral). In ventral lying, the sow’s belly region is in contact with the floor. In lateral lying, one side of the body is in contact with the floor and legs are extended to one side.

**Table 2 animals-09-00369-t002:** Effects of enrichment treatment and day of observation on time spent (proportion of observations) and number of sows (proportion of sows) in contact with enrichment, less than 1 m or greater than 1 m from enrichment.

Behaviour *		Treatment			*p* Value	
	Constant	Rotate	Stimulus	SEM	Treatment	Day **
Contacting enrichment:						
Time (prop. of obs.)	0.569	0.598	0.612	0.095	0.966	0.329
No. of sows (prop.)	0.112 ^a^	0.200 ^b^	0.182 ^b^	0.019	0.041	0.001
Less than 1M:						
Time (prop. of obs.)	0.527	0.642	0.651	0.062	0.238	0.267
No. of sows (prop.)	0.123	0.168	0.159	0.028	0.371	0.388
Greater than 1M:						
Time (prop. of obs.)	0.999	0.988	0.988	0.001	0.358	0.299
No. of sows (prop.)	0.667 ^a^	0.541 ^b^	0.532 ^b^	0.020	0.001	0.001

^ab^ LSMeans with different a superscript within the same row are significantly different (*p* < 0.05). ***** Sows were observed during an 8 h period (08:00–16:00 h) on four days (1, 8, 10 and 12) using time lapse photos at 10 min intervals. ** Day = Day of observation (days 1, 8, 10 and 12).

**Table 3 animals-09-00369-t003:** Effects of enrichment treatment and day of observation on the postures of sows observed in photo scans *. LS Means of duration (proportion of observations) and number of sows in each posture (proportion of sows).

Behaviour *	Treatment					*p*-Value	
	Control	Constant	Rotate	Stimulus	SEM	Treatment	Day **
Standing:							
Time (prop. of obs.)	0.965	0.944	0.938	0.932	0.022	0.936	0.896
No. of sows (prop.)	0.444	0.466	0.513	0.513	0.022	0.081	0.061
Sitting:							
Time (prop. of obs.)	0.283 ^a^	0.307 ^a^	0.219 ^a^	0.181 ^b^	0.041	0.035	0.201
No. of sows (prop.)	0.057	0.069	0.058	0.061	0.009	0.352	0.547
Lying:							
Time (prop. of obs.)	0.954	0.970	0.940	0.939	0.012	0.193	0.027
No. of sows (prop.)	0.343	0.314	0.298	0.294	0.023	0.430	0.492

* Sows were observed during an 8 h period (08:00–16:00 h) on four days (1, 8, 10 and 12) using time lapse photos at 10 min intervals. ** Day = day of observation (days 1, 8, 10 and 12). ^ab^ LSMeans with a different superscript within the same row are significantly different (*p* < 0.05).

**Table 4 animals-09-00369-t004:** Effects of social status and day of observation on time spent (proportion of observations) and number of sows (proportion of sows) in contact with enrichment, less than 1 m or greater than 1 m from enrichment.

Behaviour *	Social Status **			*p*-Value ***	
	Dom	Sub	SEM	SS	Day
Enrichment contact					
Time (prop. of obs.)	0.279	0.314	0.024	0.172	0.001
No. of sows (prop.)	0.386	0.397	0.009	0.939	0.001
Less than 1 m					
Time (prop. of obs.)	0.137 ^a^	0.203 ^b^	0.015	0.001	0.001
No. of sows (prop.)	0.371	0.371	0.009	0.992	0.001
Greater than 1 m					
Time (prop. of obs.)	0.854 ^a^	0.905 ^b^	0.016	0.001	<0.001
No. of sows (prop.)	0.563 ^a^	0.591 ^b^	0.013	0.042	0.001

^ab^ LS Means with a different superscript within the same row are significantly different (*p* < 0.05). * Sows were observed during an 8 h period (08:00–16:00 h) on four days (1, 8, 10 and 12) using time lapse photos at 10 min intervals. ** Social Status LS Means: Dom = Dominant; Sub = Subordinate. *** SS = Social Status, Day = Day of observation (1, 8, 10 and 12).

**Table 5 animals-09-00369-t005:** Effects of social status enrichment treatment and day on focal sow postures observed in photo scans. LS Means of duration (proportion of observations) and number of sows (proportion of sows) in each posture.

Behaviour *	Social Status **				*p*-Value	
	Dom	Sub	SEM	SS	Treat.	Day
Standing						
Time (prop. of obs.)	0.771 ^a^	0.824 ^b^	0.022	0.002	0.178	0.728
No. of sows (prop.)	0.561 ^a^	0.585 ^b^	0.013	0.007	0.023	0.407
Sitting						
Time (prop. of obs.)	0.055	0.059	0.004	0.619	0.299	0.977
No. of sows (prop.)	0.342	0.334	0.041	0.233	0.674	0.348
Lying						
Time (prop. of obs.)	0.489	0.512	0.040	0.409	0.003	0.522
No. of sows (prop.)	0.447	0.437	0.015	0.440	0.002	0.899

^ab^ LS Means with a different superscript within the same row are significantly different (*p* < 0.05). * Sows were observed during an 8 h period (08:00–16:00 h) on four days (1, 8, 10 and 12) using time lapse photos of 10 min intervals. ** Social Status LS Means: Dom = Dominant; Sub = Subordinate.

**Table 6 animals-09-00369-t006:** Effects of social status on skin lesion scores in focal sows. Sows measured on treatment days 1 and 12.

	Social Status *			
Lesion Scores **	Dom	Sub	SEM	*p*-Value
Day 1 Total	2.74 ^a^	4.46 ^b^	0.44	0.004
Shoulder Day 1	0.66 ^a^	1.17 ^b^	0.12	0.001
Day 12 Total	3.61	4.24	0.36	0.293
Shoulder Day 12	0.72	0.92	0.12	0.214

* Social Status LS Means: Dom = Dominant; Sub = Subordinate. ** Lesion scores were done using a scale of zero to three (0 = no injury, and 3 = severe injury) on 11 regions on the right and left side of sows. Total = sum of lesions for all 22 body regions. ^ab^ LS Means with a different superscript within the same row are significantly different (*p* < 0.05).

**Table 7 animals-09-00369-t007:** Effects of enrichment treatments on skin lesion scores in focal sows. Lesion scores were measured on day 1 (before enrichment) and day 12 of each treatment period.

Lesion Scores *	Treatments					
	Control	Constant	Rotate	Stimulus	SEM	*p*-Value
Day 1 Total	3.54 ^a^	2.50 ^a^	3.32 ^a^	5.41 ^b^	0.57	0.002
Shoulder Day 1	1.11 ^a^	0.47 ^ab^	0.83 ^ab^	1.40 ^c^	0.18	0.001
Side Day 1	0.63 ^a^	0.33 ^a^	0.58 ^a^	0.91 ^b^	0.13	0.028
Day 12 Total	4.01	3.42	4.50	4.11	0.27	0.446
Shoulder Day 12	0.77	0.63	1.00	0.80	0.16	0.465
Side Day 12	0.40	0.46	0.51	0.41	0.11	0.908

^ab^ LS Means with a different superscript within the same row are significantly different (*p* < 0.05). * Lesion scores were done using a scale of zero to three (0 = no injury, and 3 = severe injury) on 11 regions on the right and left side of sows. Total = sum of lesions for all 22 body regions.

**Table 8 animals-09-00369-t008:** Effect of social status and sample time point on salivary cortisol levels (LS Means, µg/dL) in focal sows.

Factor	Cortisol Concentration	SEM	*p*-Value
Social Status:			
Dominant	0.25 ^a^	0.04	0.002
Subordinate	0.43 ^b^	0.04	
Sample time point:			
Early gestation (week 5)	0.22 ^a^	0.04	0.005
Mid gestation (week 9)	0.39 ^a^	0.05	
Late gestation (week 14)	0.43 ^b^	0.04	

^ab^ LS Means with different superscripts within a factor are significantly different (*p* < 0.05).

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
