# Peer review of "Effects of Enrichment Type, Presentation and Social Status on Enrichment Use and Behaviour of Sows with Electronic Sow Feeding"

_animals, 2019, doi:10.3390/ani9060369_

Round 1
Reviewer 1 Report
This is an important study, well designed, implemented, analysed and written. I have a few minor points, detailed line by line below. The main points I think need addressing are more clarity on how the data was prepared and analysed, which will make the results, mainly the behavioural results clearer, as they are currently a little confusing (e.g. % obs vs % sows). It would be really useful to includes photos and/or a schematic diagram of the pen set up including enrichment placement. I’d also like to see the common barriers to enrichment use; biosecurity and manure handling referred back to in the discussion after mentioning only once in the summary. Since this has great practical application, these details are useful and important.
Simple Summary
L22 – suggest adding what the associate stimulus is in brackets so its clear here to the reader, e.g. (bell or whistle).
Abstract
L35 – as above add what the stimulus is, as a reader, I was curious to know here, e.g. (bell or whistle).
L37 – is that 12 days for each enrichment treatment?
L39 – if word count allows, suggest adding: for 8h (between 8am and 4pm) and 4 days/treatment (d 1, 8, 10 and 12).
L44-46 – after this line, I’d like to see a similar concluding statement to the simple summary, e.g. that enrichment is valued, therefore important and the right amount of is needed to avoid competition over access.
Introduction
L62 – “and oral stereotypies” could be added.
L63-68 – mention the manure system as a barrier, but what about the biosecurity mentioned in the summary? This is really a ‘false objection’ in my mind, but worth mentioning the rationale behind it.
L88 – perhaps expand a little on the properties of enrichment that make it appealing to pigs, e.g. chewable, deformable, destructible, edible (especially for feed restricted sows who benefit from additional sources of fibre).
Methods
L106-113 – would benefit from a little more information on the two pen designs, including the total pen dimensions (length x width), did the pens have partitioned ‘bed room’ areas, the location(s) and of the ESF and the number and location of water point(s). If easier, could add a photo or schematic of the designs and indicate locations of the enrichments too. Also, useful to know the sow genetics. These aspects can influence sow aggression, so useful for the reader to know.
L126-136 – excellent description of the detail of enrichment provision! A photo and/or schematic as mentioned above would really help too.
L150 – ‘was’ to ‘were’.
L169 – ‘datasheet’ to datasheets’.
Figure 2 – really picky so feel free to ignore, but the image is a boar, a sow might be better ?!
L201-207 – happy not to see great detail of the process, as it’s likely the kit instructions were followed. Would be useful to know whether exact instructions were followed or any adaptations to the process were made? Also missing a couple of things to enable replication of the experiment, including: were the samples analysed in duplicate or triplicate, 20 standards were run in total, but not clear how many plates and how many standards per 96-well plate?
L219 – ‘Model fitness’ to ‘Model fit’.
L211-230 – why not include day of gestation? If this was balanced, then perhaps mention that is why it was not included in the model, or is this accounted for in the sow group replicate in the random model? Also where is Sub/Dom in the behaviour model and day of observation? A bit more detail on the stats is needed to make clearer.
Having read and been confused by the results, suggest renaming the ‘Statistical Analysis’ section ‘Data Manipulation and Analysis’ with each paragraph containing more detail on how the different data types were prepared and then analysed. This way, all the detail is in one place before reading the results.
Results
Table 2 – this is a little confusing? The title mentions sows in contact and close to enrichment, but also includes greater than 1m in the table. I can’t quite follow why the time for >1m can be close to or 100% when the % for close or in contact are high?
Figure 3 – Why not have 2 bars for each with stimulus and rotate so the reader can see they are similar?
Table 3-6 – again, confusing why these don’t add up to 100%? It could be clearer if the stats methods included details of the data manipulation on how the % of obs and % of sows was calculated. I genuinely can’t work it out at the moment.
I’d like to know the parities of the Sub and Dom sows somewhere?
Discussion
L353-361 - It’s worth mentioning whether the straw posed biosecurity or manure handling issues in this study as referred to in the simple summary and manure handling only in the introduction. This was a concern for many European producers, who found it not to be as big an issue as anticipated, by ensuring good quality straw and/or chopped straw, also whether the sows mostly ate it (rather than it falling down the slats) and the benefits of additional fibre for sows potentially outweighing these issues.
L448-462 – for the cortisol results, it could be a difference in feeding time, given sows here are fed with ESF. For Sub sows who may feed at different times to Dom sows (e.g. less preferred times), they could have a slightly different diurnal pattern of cortisol and this could be contributing to the difference since saliva was only sampled at one time point.
L431-447 – On aggression, the simple summary and abstract concludes on this point, but in the results and discussion, its clear that lesions scoring were pretty low (if summed for 22 body regions the score could be a total of 66 but results show between 2 and 6!). If I were reading just these summary/abstract I would focus on aggression being an issue in relation to enrichment. But, given how much the sows used it, particularly the straw and rotation of enrichments, I would conclude that the value of enrichments would outweigh any potential added aggression over them, which could be easily minimised with appropriate volume and placement of enrichment items.

Author Response
We thank Reviewer 1 for their feedback and suggestions. Hopefully the revisions described here will clarify the study methods and address their concerns.
Simple Summary
L22 – suggest adding what the associate stimulus is in brackets so its clear here to the reader, e.g. (bell or whistle).
The suggested change has been made and more detail has been added in line 138.
“Both the bell and whistle were used at different times. The associative stimulus used was switched half-way through the study, so that no sows that returning to the study would be familiar with the stimulus.”
Abstract
L35 – as above add what the stimulus is, as a reader, I was curious to know here, e.g. (bell or whistle).
Necessary changes made.
L37 – is that 12 days for each enrichment treatment?
Yes. This has been is clarified on line 37, in the sentence describing treatments.
L39 – if word count allows, suggest adding: for 8h (between 8am and 4pm) and 4 days/treatment (d 1, 8, 10 and 12).
The suggested changes have been made.
L44-46 – after this line, I’d like to see a similar concluding statement to the simple summary, e.g. that enrichment is valued, therefore important and the right amount of is needed to avoid competition over access.
The concluding sentence has been revised. However, because the relationship between the amount of enrichment given and aggression was not studied directly the suggested wording was not used.
Introduction
L62 – “and oral stereotypies” could be added.
Agreed.
L63-68 – mention the manure system as a barrier, but what about the biosecurity mentioned in the summary? This is really a ‘false objection’ in my mind, but worth mentioning the rationale behind it.
Necessary changes made. The sentence, “Other concerns related to substrate provision include cost and biosecurity risk.” has been added.
L88 – perhaps expand a little on the properties of enrichment that make it appealing to pigs, e.g. chewable, deformable, destructible, edible (especially for feed restricted sows who benefit from additional sources of fibre).
A sentence has been added describing the properties of effective enrichment- but note that the content is similar to line 75.
Methods
L106-113 – would benefit from a little more information on the two pen designs, including the total pen dimensions (length x width), did the pens have partitioned ‘bed room’ areas, the location(s) and of the ESF and the number and location of water point(s). If easier, could add a photo or schematic of the designs and indicate locations of the enrichments too. Also, useful to know the sow genetics. These aspects can influence sow aggression, so useful for the reader to know.
The layout of the pens has been added in Figure 1, and the genetics supplier is named in section 2.1.
L126-136 – excellent description of the detail of enrichment provision! A photo and/or schematic as mentioned above would really help too.
Pictures of straw, rope and wood enrichments have been added (Figure 2).
L150 – ‘was’ to ‘were’.
The word ‘was’ has been left as it refers to the sub-sample.
L169 – ‘datasheet’ to datasheets’.
Necessary changes have been made.
Figure 2 – really picky so feel free to ignore, but the image is a boar, a sow might be better ?!
The diagram (now Figure 4) has been edited to represent a sow.
L201-207 – happy not to see great detail of the process, as it’s likely the kit instructions were followed. Would be useful to know whether exact instructions were followed or any adaptations to the process were made? Also missing a couple of things to enable replication of the experiment, including: were the samples analysed in duplicate or triplicate, 20 standards were run in total, but not clear how many plates and how many standards per 96-well plate?
Changes have been made indicating that the manufacturer’s instructions were followed, the number of control and unknown samples on each plate is listed and all samples were run in duplicate.
L219 – ‘Model fitness’ to ‘Model fit’.
The requested change has been made.
L211-230 – why not include day of gestation? If this was balanced, then perhaps mention that is why it was not included in the model, or is this accounted for in the sow group replicate in the random model? Also where is Sub/Dom in the behaviour model and day of observation? A bit more detail on the stats is needed to make clearer.
A sentence has been added, “Day of gestation was controlled as each group (pen of sows) received all treatments, and the order of enrichment treatments was randomized for each group.”
A sentence describing the model used for focal sows which included social status has been added at line 249. Day of observation is included in the model described on line 248, as follows: “Fixed effects in the group behaviour model included enrichment treatment, day of treatment (day 1, 8, 10 and 12) and their interaction.” The specific days have been added.
Having read and been confused by the results, suggest renaming the ‘Statistical Analysis’ section ‘Data Manipulation and Analysis’ with each paragraph containing more detail on how the different data types were prepared and then analysed. This way, all the detail is in one place before reading the results.
The ‘Statistical Analysis’ section has been divided into two sections, with one describing data manipulation and the other, analysis. More information has been added on the analysis methods.
Results
Table 2 – this is a little confusing? The title mentions sows in contact and close to enrichment, but also includes greater than 1m in the table. I can’t quite follow why the time for >1m can be close to or 100% when the % for close or in contact are high?
The title has been changed to include greater than 1 M observations. Regarding the percentages:
The time spent in each position is the percent of observations where the behavior was observed, based on the entire group of sows (or group of focal sows). The total is greater than 100% because at any one time some sows may be in contact with enrichment, some within 1M and some >1M. For example, if there was one sow doing each behaviour in all photos, the time spent would all be 100%. That is why we also included information on the number of sows (%) doing each behaviour, which indicates the average number of individuals performing the behavior when it was observed. We suggest the two measures together give a better description of the prevalence of the behaviour than either measure alone.
Time spent = the percentage of scans or time point observations where the particular behavior (contact, <1 meter, or >1 meter) was observed
Number of sows = average no of animals (% of group) observed in a particular behavior when it was observed
Figure 3 – Why not have 2 bars for each with stimulus and rotate so the reader can see they are similar?
A more elaborate graph has been added showing separate enrichment interactions for the Constant, Rotate and Stimulus treatments.
Table 3-6 – again, confusing why these don’t add up to 100%? It could be clearer if the stats methods included details of the data manipulation on how the % of obs and % of sows was calculated. I genuinely can’t work it out at the moment.
Hopefully the changes and discussion above provide some clarity.
I’d like to know the parities of the Sub and Dom sows somewhere?
This information has been added to results section 3.2. “The parity distribution of dominant and subordinate focal sows was as follows: Dominants- parity 2: 39%; 3:11%, 4:17%, 5:17%, 6: 11% and 7: 5% and Subordinates- Parity 1:11%, 2: 28%, 3: 22%, 4: 22%, and 5:16% (18 Dom and 18 Sub sows in total).”
Discussion
L353-361 - It’s worth mentioning whether the straw posed biosecurity or manure handling issues in this study as referred to in the simple summary and manure handling only in the introduction. This was a concern for many European producers, who found it not to be as big an issue as anticipated, by ensuring good quality straw and/or chopped straw, also whether the sows mostly ate it (rather than it falling down the slats) and the benefits of additional fibre for sows potentially outweighing these issues.
A statement regarding the lack of problems when straw was provided in partially slatted pens has been added. “This study also showed that providing 300 grams of straw per sow per week in partially slatted pens did not cause any blockage of the liquid manure system. Most of the material was consumed by sows and only a small amount entered the pits.”
L448-462 – for the cortisol results, it could be a difference in feeding time, given sows here are fed with ESF. For Sub sows who may feed at different times to Dom sows (e.g. less preferred times), they could have a slightly different diurnal pattern of cortisol and this could be contributing to the difference since saliva was only sampled at one time point.
This is an interesting suggestion; the information has been added as follows, “Another possibility may be that due to ESF feeding (here with a 9am reset time), the diurnal cortisol rhythm of subordinate sows may be shifted. Further research to determine whether the ESF system affects the circadian rhythm of dominant and subordinate sows differently would be of interest.”
L431-447 – On aggression, the simple summary and abstract concludes on this point, but in the results and discussion, its clear that lesions scoring were pretty low (if summed for 22 body regions the score could be a total of 66 but results show between 2 and 6!). If I were reading just these summary/abstract I would focus on aggression being an issue in relation to enrichment. But, given how much the sows used it, particularly the straw and rotation of enrichments, I would conclude that the value of enrichments would outweigh any potential added aggression over them, which could be easily minimised with appropriate volume and placement of enrichment items.
No changes were made as the low level of lesion scores has already been noted (line 486).
Reviewer 2 Report
Manuscript ID: animals-470692
Type of manuscript: Article
Title: Effects of enrichment type, presentation and social status on enrichment use and behaviour of sows with electronic sow feeding
Authors: Cyril Roy, Lindsey Lippens, Victoria Kyeiwaa, Yolande Seddon, Laurie
Connor, Jennifer Brown *
General:
This study evaluated the impact of point source enrichment on group-housed sow behavior. While the justification for looking into enrichment items which are compatible with swine facilities that use a flush manure system, this study was unable to provide a large enough sample size to support the claims which are made in the discussion. It appears that six pens of 20 sows were repeatedly observed for behavioral response to three enrichment treatments (a total of 120 sows in this study). Given that the pen is the experimental unit, this gives a sample size of 6. A complex generalized linear mixed model was run across this sample size; which is very concerning. In addition, the outcome variable of “duration” is not a continuous sampling of behavior duration, but a proportion of scans in which the sows were observed to be in specific postures or distance from enrichment. Therefore, any reference to duration of behavior is incorrect. Another very concerning issue is the inability of the authors to tease apart lesion scores given that they were collected in succession. Lesions on “day 1” for the third treatment is actually “day 26” for the first treatment; therefore, these measurements are not independent.
The authors should include the limitation that their feed competition test may not have evaluated social dominance, but instead, correctly evaluated individual differences in food motivation. This would explain why sows which “won” the test were also more interested in the ESF feeder. Finally, a major oversight from this manuscript is the lack of comparison to a nearly identical study published by Horback et al. in 2016. The authors should incorporate this study in their interpretation of their results.
Introduction
Line 54: Place comma after the word “”members”
Line 57: Are the authors stating that enrichment can increase both positive and negative social interaction?
Line 74: How did the study evaluate/determine “reduced weaning stress” due to conditioned stimulus presentation?
Line 74-77: These statements on dominance and enrichment use could possibly be placed earlier (after/before lines 61-62?) when the authors vaguely refer to negative aspects of group housing sows.
Methods
Lines 108-109: Were the study pens evenly split across these two types of pen designs?
Line 117: To clarify, could the word “group” be replaced with “pen”; each enrichment was provided to each pen of sows.
Line 117-118: What is the justification for 12 days of enrichment provision per treatment? Why only 2 days “down” in-between treatments?
Line 126-136: How was is guaranteed that the sows from neighboring pens did not hear the conditioned acoustic stimulus? And how long and at what decibels was the sound amplified?
Figure 1. Can the acoustic stimulus be illustrated in some way in the figure to show how stimulus treatment is different than rotation treatment (like an asterisk before each enrichment label)?
Line 152: What was the metric used to determine that aggression associated with mixing had been resolved? Less than 5 fights per hour or day? Was this actually measured? Three to five days is a large difference in terms of sows establishing hierarchy.
Lines 151-159: Can the authors provide external support for using feed competition tests to determine social dominance? How can they rule out that latency to consume limited feed was not due to difference in feed motivation, rather than social rank? Authors should clarify that this feed test was among 20 +/- sows in each pen, correct?
Line 161-162: Please include the total number of photos analyzed for clarity. One photo taken every 10 minutes for 8 hours = 6 photos per hour x 8 hours = 48 photos/day; correct? For a total of 192 photos per pen over the 4 days of observation. This provides baseline for proportion data given later (i.e., 50% of scans the sows were observed in contact with enrichment = 96 scans)
Lines 162-163: Was there no inter-observer reliability for behavior data collection from still photos? What about intra-observer reliability?
Line 162: Why was data collected during this time period and on these particular days?
Lines 176-184: What about inter-rater reliability for lesion scoring? This is a very contentious topic and could be incredibly subjective.
Line 192: Do the authors mean that 3 dom and 3 sub saliva collection per “pen” and not “group”? If so, it is suggested to switch the words to clarify.
Lines 212-215: What the authors are describing is not a duration, it is a proportion. Please correct this sentence.
Lines 217-221: Mixed regression model was used to analyze behavior, however, this study has a sample size of 6; is that correct? Six pens of 20+/-2 sows? How can the authors justify running this statistical test on such a small sample size?
Results
Figure 2. What is the significance level for the annotated letters? P<0.05, or <0.01? And why are only these particular time x treatment data provided?
Lines 162 and 259: Inconsistency in formatting of time: (e.g., “ 4 pm” and “16:00 hr”)
Lines 310-315: How are the authors justifying repeated measures of lesions when previous wounds acquired in rotation treatment could still be healing during the stimulus treatment. This lesion analysis is very flawed; one cannot separate lesions given that “day 1” for the third treatment is essentially “day 26” for the study (or “day 27-30” post-mixing). While “day 1” for the first treatment is actually “day 4-6” post-mixing.
Table 7. Is this the average lesion score per dom or sub sow from the four “day 1” per treatment and the four “day 12 per treatment?
Line 334: A single parenthesis after “hr” needs to be removed.
Discussion
Line 342: Remove the commas after “”wood)”
Line 343-344: This is the first mention of animal preference as an objective for this study. Why was preference not mentioned in the introduction? This is also not “proof” of preference, as the animals did not have a choice of all treatments at the same time.
Lines 463-465: These two sentences should be included at the end of the previous paragraph rather than have a 2 sentence paragraph.
Line 353-361: The authors need to include the Horback, Pierdon & Parsons (2016) paper which specifically looked at behavioral preference for rope, wood and rubber enrichment items in group-housed sows. This is a major oversight in the literature review for this manuscript.
(Horback, K. M., Pierdon, M. K., & Parsons, T. D. (2016). Behavioral preference for different enrichment objects in a commercial sow herd. Applied animal behaviour science, 184, 7-15.)
Lines 409-403: The authors should include the limitation that their feed competition test may not have evaluated social dominance, but instead, correctly evaluated individual differences in food motivation. This would explain why sows which “won” the test were also more interested in the ESF feeder.
Author Response
We thank Reviewer 2 for their feedback and suggestions. Hopefully the revisions described here will clarify the study methods and address their concerns.
Line 54: Place comma after the word “”members”.
The change has been made.
Line 57: Are the authors stating that enrichment can increase both positive and negative social interaction?
The statement refers to group housing with social enrichment only. Increased social interaction can have positive or negative effects depending on social status within the group and other management factors.
Line 74: How did the study evaluate/determine “reduced weaning stress” due to conditioned stimulus presentation?
In the Dudink et al (2006) study, aggression and lesion scores were lower following weaning compared to controls. Necessary changes made.
Line 74-77: These statements on dominance and enrichment use could possibly be placed earlier (after/before lines 61-62?) when the authors vaguely refer to negative aspects of group housing sows.
The introductory paragraph is about group housing in general, not about enrichment. In Canada enrichment was required as of 2014, but is not yet provided on all farms. In the USA there are no requirements for enrichment and few barns provide it. Based on the Canadian code of practice, group housing itself is considered a form of enrichment, however, we indicate that social grouping by itself may not benefit sows, particularly subordinate individuals.
Methods
Lines 108-109: Were the study pens evenly split across these two types of pen designs?
Yes, the study was evenly split. There were three replicates in each pen design.
Line 117: To clarify, could the word “group” be replaced with “pen”; each enrichment was provided to each pen of sows.
The word pen has been added for clarity.
Line 117-118: What is the justification for 12 days of enrichment provision per treatment? Why only 2 days “down” in-between treatments?
The 12 day treatment (14 day cycle) allowed us to provide each sow group with 4 treatments over an eight week period, starting at 5 weeks gestation. This resulted in sows being on trial until week 14 of gestation, at which point they were approaching parturition. The two day down time between treatments allowed for cleaning of enrichments and preparation for the next treatment.
Line 126-136: How was it guaranteed that the sows from neighboring pens did not hear the conditioned acoustic stimulus? And how long and at what decibels was the sound amplified?
Due to the design of the gestation room, sows in neighboring pens could hear the stimulus, but it was not paired with enrichment provision. When the Stimulus treatment was given, the other pens in the room were given the Control or Constant treatment so they had no association between the stimulus and management practices. Each stimulus lasted approximately 2 seconds and was clearly audible. The stimulus duration has been added to section 2.2.
Figure 1. Can the acoustic stimulus be illustrated in some way in the figure to show how stimulus treatment is different than rotation treatment (like an asterisk before each enrichment label)?
The Rotate and Stimulus treatments were the same, except for the use of the stimulus. We do not know of any clear way to revise this figure to indicate the difference.
Line 152: What was the metric used to determine that aggression associated with mixing had been resolved? Less than 5 fights per hour or day? Was this actually measured? Three to five days is a large difference in terms of sows establishing hierarchy.
The feed competition test was carried out on all three days for all groups. Feed was provided on days 3 and 4 after mixing, and the final selection of focal animals was made on day 5.
Lines 151-159: Can the authors provide external support for using feed competition tests to determine social dominance? How can they rule out that latency to consume limited feed was not due to difference in feed motivation, rather than social rank? Authors should clarify that this feed test was among 20 +/- sows in each pen, correct?
The feed competition test was adapted from the study by Anderson et al (1999). The reference has been added. A study by Brouns and Edwards (1994) found that rank order in pair-wise feeding competition tests was highly correlated with rank order determined by observing social interactions. In general all gestating sows are highly food motivated due to feed restriction, although it will vary. Recent unpublished work from our group indicates stronger food motivation in older/larger sows compared to gilts.
Yes. The feed competition test was between the 20 +/-2 sows. Gilts were present but were not selected as focal animals. Additional wording to this effect has been added.
Line 161-162: Please include the total number of photos analyzed for clarity. One photo taken every 10 minutes for 8 hours = 6 photos per hour x 8 hours = 48 photos/day; correct? For a total of 192 photos per pen over the 4 days of observation. This provides baseline for proportion data given later (i.e., 50% of scans the sows were observed in contact with enrichment = 96 scans)
Correct. More detail on the number of observations has been added in Section 2.3.
Lines 162-163: Was there no inter-observer reliability for behavior data collection from still photos? What about intra-observer reliability?
No inter-observer reliability scores were calculated as data transcription was done by one person. Intra-observer reliability was not assessed.
Line 162: Why was data collected during this time period and on these particular days?
The 8 h data collection captured most of the day’s activity and commenced at 8am following feeder reset at 9am. The 8am start thus avoided the initial high level of activity which occurred at 9am. Also, this study followed the same procedures as a parallel study in free-access stalls, where all sows were fed at 7am, so the 8am start time coincided with completion of feeding in the free-access system. Days 1, 8, 10 and 12 were selected as a representative sample of time points that included all enrichment types and the full treatment time span.
Lines 176-184: What about inter-rater reliability for lesion scoring? This is a very contentious topic and could be incredibly subjective.
Lesion scoring was done by two trained technicians. The technicians trained together however, no reliability estimates were obtained.
Line 192: Do the authors mean that 3 dom and 3 sub saliva collection per “pen” and not “group”? If so, it is suggested to switch the words to clarify.
We do not understand the question.
Lines 212-215: What the authors are describing is not a duration, it is a proportion. Please correct this sentence.
The sentence has been corrected.
Lines 217-221: Mixed regression model was used to analyze behavior, however, this study has a sample size of 6; is that correct? Six pens of 20+/-2 sows? How can the authors justify running this statistical test on such a small sample size?
Each pen received each treatment consecutively, so the sample size is 6 x 4= 24.
Results
Figure 2. What is the significance level for the annotated letters? P<0.05, or <0.01? And why are only these particular time x treatment data provided?
A more detailed figure has been provided based on comments from Reviewer 1. Differences are at P <0.05 level. In the model, enrichment contact by sows was significantly affected by treatment and day. This figure was provided to explain the relationship.
Lines 162 and 259: Inconsistency in formatting of time: (e.g., “ 4 pm” and “16:00 hr”)
Time measures have been revised for consistency.
Lines 310-315: How are the authors justifying repeated measures of lesions when previous wounds acquired in rotation treatment could still be healing during the stimulus treatment. This lesion analysis is very flawed; one cannot separate lesions given that “day 1” for the third treatment is essentially “day 26” for the study (or “day 27-30” post-mixing). While “day 1” for the first treatment is actually “day 4-6” post-mixing.
Treatment order was randomized for each group to control for carry-over effects, and the majority of lesions were very mild (scores of 0 and 1 predominantly). Therefore healing in most cases can be expected within a matter of days.
Table 7. Is this the average lesion score per dom or sub sow from the four “day 1” per treatment and the four “day 12 per treatment?
No. The individual measures for each day were used in the analysis and sow was included in the model as a repeated measure. The model used for lesions has been clarified in the methods.
Line 334: A single parenthesis after “hr” needs to be removed.
The requested change has been made.
Discussion
Line 342: Remove the commas after “”wood)”
Change made.
Line 343-344: This is the first mention of animal preference as an objective for this study. Why was preference not mentioned in the introduction? This is also not “proof” of preference, as the animals did not have a choice of all treatments at the same time.
We agree. The objective was not to evaluate enrichment preferences per se, but to compare the various treatments. The sentence has been revised as follows, “The results indicate that sows interacted with straw, rope and wood in decreasing order.”
Lines 463-465: These two sentences should be included at the end of the previous paragraph rather than have a 2 sentence paragraph.
Changes made.
Line 353-361: The authors need to include the Horback, Pierdon & Parsons (2016) paper which specifically looked at behavioral preference for rope, wood and rubber enrichment items in group-housed sows. This is a major oversight in the literature review for this manuscript.
(Horback, K. M., Pierdon, M. K., & Parsons, T. D. (2016). Behavioral preference for different enrichment objects in a commercial sow herd. Applied animal behaviour science, 184, 7-15.)
Thank you for the suggestion. The reference has been added in appropriate places.
Lines 409-403: The authors should include the limitation that their feed competition test may not have evaluated social dominance, but instead, correctly evaluated individual differences in food motivation. This would explain why sows which “won” the test were also more interested in the ESF feeder.
Sows in general are highly food motivated due to feed restriction. Feed competition tests have been used by other researchers to assess social rank based on the assumption that feed is a valued resource and that dominant individuals will control access to it. The Anderson et al (1999) reference has been added to the methods section.
We agree there may be some confounding with feed motivation, and have added the sentence, “Furthermore, because social status was determined based on a feed competition test, it may be that differences in social status are confounded with feeding motivation.”
Reviewer 3 Report
Review of manuscript Effects of enrichment type, presentation and social status on enrichment use and behaviour of sows with electronic sow feeding (Animals-470692)
Note: I assume that the track changes in the manuscript are due to an earlier revision.
General remarks:
This manuscript presents findings on the effectiveness of enrichment type, presentation and social status on enrichment use and behaviour in sows. I think this an interesting area of study that requires attention.
However, I feel there is room for improvement. The authors investigated and presented many factors but as such I lost track of the overall aim of the study at times. I believe the issue is a matter of structuring and leading the reader through the different parts of the study.
General comments:
The way I see this study the authors aimed to investigate three aspects: 1) investigate the enrichment presentation (constant, rotate, stimulus); 2) investigate which enrichment type is most effective within the rotate/stimulus treatments; and 3) investigate how social status plays a role in enrichment use. I would encourage the authors to go through the manuscript and really make sure that that train of thought is clear and now at times it was hard to follow why certain results were presented.
My main concern is with the second aim for a few reasons. First of the order of the EE provided in these two treatments always followed Rope – Straw – Wood from my understanding and I think this could have influenced the sows’ responses. Additionally, the photos to assess EE interaction were taken on specific days and so you would always have 2x as many photos for the Rope compared to Straw and Wood. This is also noticeable in Figure 5 where you then do not have results for day 3 (Straw) and day 5 (Wood). I do believe it is an interesting aside to investigate but I am not sure if this design would be the most appropriate and would suggest to remove it or make sure its limitations are be acknowledged.
Additionally, I have some doubts if the photographs at 10 min intervals would be enough to really pick up interaction with the EE (interactions can be short). But also just standing on the EE / in contact with the EE simply by chance could happen, especially, in the case of the straw treatment as it spread out over the solid floor. Though I do feel the authors were cautious in their ethogram description of ‘contacting’ but it is something to consider.
For the first aim, regarding the Stimulus treatment the authors acknowledge the fact that the sows were not trained and so they cannot be sure if the sows associated the stimulus with the arrival of enrichment in the discussion. I was expecting to see information on the training in the material and methods section and wonder why no training was used? If the sows did not associate the stimulus with the EE then does not undermine the reason for the Stimulus treatment in the first place?
Finally, there is a lot of information presented in the manuscript and I feel that it might be worthwhile to consider removing the behavioural postures. With the current description I do not feel that this was part of the main aim of the study and it is relatively little discussed. Removing this aspect might help increase the focus and clarity of the paper.
More specific comments regarding the manuscript follow below.
Simple summary/abstract:
Based on the general comments some rewriting/refocussing of the abstracts might be required.
L22-25: if contact and time spent in different postures was measured in all sows, I do not see the need to point out it was also measured on a subset of dominant and subordinate sows (they would be included in all sows right). I am more interested if the skin lesions/saliva samples were taken from all sows or the subset?
Introduction:
The introduction is concise and generally well written. I only have a few minor comments.
L90-91: I believe this sentence is added after initial revision, but I am not sure of its location. I think it is because I do not see the flow from L87 to L88-90 and then the concluding sentence in L91-93; this needs some rephrasing for the authors to really make their point. Also, this sentence appears to be different font/size?
L96: ‘the trial’.
Material and methods
See general comments.
There were two pen designs used in the study; do you think this could have influenced the results?
I am also curious about the inclusion of the ‘no enrichment’ treatment as in the European Union enrichment is required (though compliance can be discussed). Would it not be better to go from the constant as the supposed ‘standard’?
Considering that there were days in between the different EE, I assume that any left-over straw after a straw EE day was removed (if there was any left or perhaps it was the opposite and it was actually not lasting the full day)?
L138-140: ‘sows that were returning to the study’ I do not follow this; why were there any sows possibly returning to the study? And why would switching the stimulus and having the sows unfamiliar with the stimulus be important (see also training comment)? Note that this sentence is repeated in L146-147.
Regarding the saliva samples, the authors pointed out in the discussion that it was not possible to determine treatment differences (L493-495), and this was my first question when reading about it in the material and methods. At that point to me it made no real sense because I was under the assumption that it was done to assess differences between treatments. If the authors make the real aim of the saliva samples/cortisol assessment clear earlier in the material and methods, you can avoid this. At the same time, it should be correcting in the statistical analysis because there is enrichment treatment mentioned as a fixed effect.
Skin lesion score were assessed for many different body parts but in the end the authors assessed a total score, head, shoulder and side. I would encourage the authors to also look at the hindquarters as we know that lesions occur more frequently posterior in the case of bullied pigs which might be important for the subordinate sows.
Results:
See general comments.
I think in part what made it difficult to track the aims and the presented results and keep them straight while reading the papers is the order in which they were presented. For example, it starts with (1) enrichment use by EE presentation treatment and the type of EE; followed by (2) postures by EE presentation treatment; then focal sows enrichment use by (3) social status and then by (4) EE presentation treatment and so on. I think for me it would be easier to follow if for example if 1 and 4 were kept together to see all the enrichment use first.
In general, it is not clear to me if interactions were tested for or found in the analysis?
L276: Delete ‘rope (day 1 and 8)’. Based on the superscripts in Figure 5 there is actually no difference between rope (day 1 and 8) and straw (day 10) in the rotate and stimulus treatment.
L300-302: Did parity have any effect on the outcomes?
L307-309: I don’t really understand what the authors are trying to say here or how they got the range; to me from Table 4 it seems it should be 36 – 59%?
L315-316: from Table 4 I cannot tell if this result is correct or how much higher sow presence was on day 10?
I assume that the results from Table 4 and 5 came from the same model which accounted for social status, treatment and day and that for presenting the results were just split into two tables. Was there any interaction between social status and treatment for enrichment use?
L351: ‘numerically higher lesion scores on all body regions’, initially I thought these results where referring to Day 1 in the table which are actually significant but on second look I assume it was the Day 12 results that were most interesting for the purpose of the paper. However, if on Day 1 before the EE treatment started the Dom and Sub sows already differed in their skin lesions did they authors analyse the results for Day 12 using the Day 1 scores as a covariate (similar for treatment results)?
L374-375: Delete, this is repetition from the material and methods.
Tables:
Table 1:
Tiny thing but I would swap the posture and location in the title so that it matches the rows of the table.
Table 2:
Change table title to ‘effects of enrichment treatment and day of observation on time spent (% of observations) and number of sows (% of sows) in contact with enrichment, less than 1M or greater than 1M from enrichment’
Use this same sort of phrasing for all relevant tables as now for example Table 4 still mentions ‘near or in contact with’.
I think it can be simplified a bit by for the sub-rows simple state time (% of observation) and No. of sows (% of sows) rather than keeping the definition of in contact, <1M and >1 M within each row.
Table 3/6:
In case the authors feel they want to keep the results I would recommend tidying up the row headers a bit by using sub-rows for each behaviour (e.g ‘Standing’) and simply name them ‘duration (% of obs)’ and ‘no. of sows (%)’.
Figures:
Figure 3: incidence/day; the recordings were analysed for 10 min every 2 hours which would have given 80 min of video per day. Is the incidence in this figure extrapolated to cover an entire day (24h?) or is it in reality day = 80 min? Again, group = treatment?
Discussion:
The discussion can probably be refined following changes made to the material and methods/results. For now, I only have a few minor comments.
Possibly some small reminder on the study aims/objectives might be useful to guide readers before starting straight into the discussion of results.
L383: there was no difference in the amount of contacting between straw and rope within the rotate/stimulus treatment (see earlier comment).
L389: the within 1 meter results were not shown in the manuscript so I would delete this part of the sentence.
L393: ‘providing 300 grams of straw per sows per week’; technically was this not provided per day (and 2 times over a 12-day period so I guess okay that almost equals per week). I guess I am just a bit cautious here as this sentence can be interpreted as that the system won’t block if you provide this much straw but I am unclear if any left-over straw was removed after the treatment day.
L415: Delete ‘with enrichment’ (repeated).
L443-451/L466-475: see earlier comment regarding maybe removing this aspect from the manuscript. The goal for including it here is more clear but one can argue if actually increasing activity is ‘normal’ as pigs are generally quite inactive.
L484-487: I do not believe you can assess this hypothesis based on the current study design; the EE rotated and lesions were only measured on day 1 (rope) and day 12 (wood) and so you cannot really make any statement on the lesions when straw was given.
L474/505: Might be interpreting the authors wrong here but I got the impression that in L474 they claim there were enough enrichments to reduce competition and here in L505 they suggest there was competition. This might need some clarification.
Conclusions
See general comments.
L527: Delete ‘being’.
Author Response
Please see comments in the attached file.

Round 2
Reviewer 2 Report
After reading the revised manuscript, it is clear that there are three remaining issues with the statistical analysis. First, with only 6 pens to assess sow behavior and posture, it is important that the authors state the degrees of freedom used for their analysis. Second, the authors are not appropriately dealing with their temporal correlation between days. They are fitting time as a fixed effect, which would control for temporal trends in average response, but they are not taking into account the correlation between temporally adjacent observations. Finally, the most significant issue with the data analysis is that the authors ran a GLMM using Poisson distribution, however, Poisson is used to model count variables, and this study is dealing with continuous proportional data (bounded between 0 and 1). The authors either need to go back to the raw counts (instead of proportion), or they needed to do a GLMM with beta regression. In the end, this study has an extremely small sample size, and therefore, conclusions made (and the manuscript title) should be edited to state that the results can not be generalized to the larger population of group-housed sows.
Author Response
In response to Reviewer 2: Behaviour measurements were done on 6 pens of sows, with four enrichment presentations (treatments) observed on four days. Therefore, for all group behavior observations, n= 6*4*4=96. However, there will be some reductions in n values for some behaviors measured. For example, “Contact with enrichment” is not possible in the control group with no enrichment. Therefore, specific N and F values have been added to P values when necessary.
Changes were made in the statistical section of manuscript to reflect that the model accounted for correlation between days (day as a repeated measure). All behavior data were changed to proportion and modeled with GLMM using beta regression with log link. The original tables have been replaced and some of the results have changed.
Reviewer 3 Report
Review of manuscript Effects of enrichment type, presentation and social status on enrichment use and behaviour of sows with electronic sow feeding (Animals-470692-revised V2)
Thank you to the authors for their explanations and work to improve the manuscript. Minor comments below.
L43: ‘contact enrichments’ – ‘in contact with enrichments’ or ‘contacting enrichments’
Fig. 5 : Capitalize ‘wood’ if possible for consistency. Glad to see Fig 5 superscripts are updated, now they indeed match the text. That said you can even be more specific:
e.g. L288-289: ‘On day 10, when straw was provided in the Rotate and Stimulus treatments, sows contacted enrichment more frequently than when rope (days 1 and 8) or wood (day 12) were provided.’ – Also more contacted than when wood (day 1,8,10,12) was provided in your Constant treatment.
L312-316: Check formatting (double spacing).
Author Response
Review of manuscript Effects of enrichment type, presentation and social status on enrichment use and behaviour of sows with electronic sow feeding (Animals-470692-revised V2)
Thank you to the authors for their explanations and work to improve the manuscript. Minor comments below.
L43: ‘contact enrichments’ – ‘in contact with enrichments’ or ‘contacting enrichments’
Changed to ‘in contact with enrichments’, as suggested.
Fig. 5 : Capitalize ‘wood’ if possible for consistency. Glad to see Fig 5 superscripts are updated, now they indeed match the text. That said you can even be more specific:
e.g. L288-289: ‘On day 10, when straw was provided in the Rotate and Stimulus treatments, sows contacted enrichment more frequently than when rope (days 1 and 8) or wood (day 12) were provided.’ – Also more contacted than when wood (day 1,8,10,12) was provided in your Constant treatment.
Wood has been capitalized in Figure 5, and the statement, ‘Also, more sows contacted the straw than when wood (day 1, 8, 10 and 12) was provided in the Constant treatment (Figure 5).’ has been added at line 290-291.
L312-316: Check formatting (double spacing).
Line spacing has been adjusted to single spacing.
Round 3
Reviewer 2 Report
The authors have made great efforts to meet nearly all suggestions to improve the manuscript. One large issue is still the matter of sample size. The authors are still overstating their results for their sample size (n = 6, six pens were observed; while the authors repeatedly say their sample size is 24, but this is repeated measures of non-mutually exclusive observations of 6 pens). I wouldn’t have as much of a problem if they’d framed this as a pilot study and been less conclusive in their conclusions.Author Response
We thank reviewer 2 for their comments. The n of 6 is now specified on lines 290 (enrichment interaction, day effect) and 314 (sitting posture, treatment effect), along with numerator and denominator degrees of freedom from the repeated measures model.